# REFUEL: Exploring Sparse Features in Deep Reinforcement Learning for Fast Disease Diagnosis

**Yu-Shao Peng**
HTC Research & Healthcare
ys_peng@htc.com

**Kai-Fu Tang**
HTC Research & Healthcare
kevin_tang@htc.com

**Hsuan-Tien Lin**
Department of CSIE,
National Taiwan University
htlin@csie.ntu.edu.tw

**Edward Y. Chang**
HTC Research & Healthcare
edward_chang@htc.com

## Abstract

This paper proposes REFUEL, a reinforcement learning method with two techniques: *reward shaping* and *feature rebuilding*, to improve the performance of online symptom checking for disease diagnosis. Reward shaping can guide the search of policy towards better directions. Feature rebuilding can guide the agent to learn correlations between features. Together, they can find symptom queries that can yield positive responses from a patient with high probability. Experimental results justify that the two techniques in REFUEL allow the symptom checker to identify the disease more rapidly and accurately.

## 1 Introduction

One of the rising needs in healthcare is self-diagnosis. About 35% of adults in the U.S. had the experience of self-diagnosing their discomfort or illness via online services, according to a survey performed in 2012 [12]. Although online searches are convenient media, search results can often be inaccurate or irrelevant [12]. The process of disease diagnosis can be considered as a sequence of queries and answers: a doctor inquires a patient of his/her symptoms to make a disease prediction. During the Q&A process, the doctor carefully chooses relevant questions to ask the patient with twin performance aims. First, each symptom probing should obtain maximal information about the patient's state. Second, after a series of Q&A, the diagnosis should be accurate.

From the machine learning perspective, the doctor's iterative decision making process can be viewed as *feature acquisition* and *classification*. The costs for feature aquisition are non-trivial in the decision making process, because acquiring a large amount of symptoms from a patient is annoying and impractical. One big family of existing methods that take the feature aquisition costs into account are tree-based. For instance, cost-sensitive decision trees were studied by Xu et al. [19] and Kusner et al. [5]; and cost-sensitive random forests by Nan et al. [7, 8] and Nan and Saligrama [6]. Recently, reinforcement learning (RL) was shown to be a promising approach to address the sequential decision problem with acquisition costs. Janisch et al. [2] use a double deep Q-network [15] to learn a query strategy. Their experimental results show that their RL approach outperforms tree-based methods.

Even thought the RL method can reach higher prediction accuracy, applying this method to the disease diagnosis task encounters two challenges. First, the numbers of possible diseases and symptoms are several hundreds, which lead to a *large search space*. Second, the number of symptoms that a patient actually suffers from is much less than that of possible symptoms, which results in a *sparse feature space*. This specialty makes the search more difficult because exploring these few symptoms in the sparse feature space is time-consuming. To alleviate the first problem, Tang et al. [14] and Kao

et al. [3] proposed ensemble models. Although the search space in each model is reduced, which resulted in improved performance, the problem of the sparse feature space remains open.

This paper proposes the REFUEL[1] model, which is a reinforcement learning model with *reward shaping* and *feature rebuilding* techniques. Reward shaping gives the agent an additional reward to guide the search towards better directions in the sparse feature space. Feature rebuilding enables the agent to discover the key symptoms by learning a better representation. Combining these two techniques, the agent can effectively learn a strategy that achieves higher prediction accuracy. Our empirical study demonstrates that our proposed techniques can productively explore in the sparse feature space and obtain high disease-prediction accuracy (e.g., achieving $91.71\%$ top-5 accuracy among 73 diseases).

## 2 Diseases Diagnosis Using Reinforcement Learning

We consider the disease diagnosis problem as a sequential decision problem. The doctor who interacts with a patient can be considered as an agent. The agent's actions in the decision process include feature acquisition and classification. Initially, the patient provides an initial symptom to the agent. Then, the agent starts its feature acquisition stage. At each time step, the agent asks the patient whether he/she suffers from a certain symptom. The patient answers the agent with true/false indicating whether he/she suffers from that symptom. After the agent has acquired sufficient symptoms, the agent switches to the classification stage and makes a disease prediction.

Suppose we have a medical dataset consisting of patients' symptoms and diseases. Let $\mathcal{X} = \{0,1\}^m$ denote the set of possible symptom profiles, where $m$ is the number of symptoms. A patient's symptom profile $x \in \mathcal{X}$ is said to have the $j^{th}$ symptom if $x_j = 1$; otherwise, it does not have $j^{th}$ symptom. We call $x_+ = \{j \colon x_j = 1\}$ and $x_- = \{j \colon x_j = 0\}$ the sets of *positive* and *negative* symptoms of $x$, respectively.[2] Also, we say that $x$ is sparse if $|x_+| \ll m$. Let $\mathcal{Y} = \{1, \ldots, n\}$ be the set of labels, where $n$ is the number of diseases, and $\mathcal{D} = \left\{ \left( x^{(j)}, y^{(j)} \right) : x^{(j)} \in \mathcal{X}, y^{(j)} \in \mathcal{Y} \right\}_{j=1}^{k}$ be the dataset of $k$ labeled examples. We say that $\mathcal{D}$ is sparse if for all $x \in \{x \in \mathcal{X} \colon (x, y) \in \mathcal{D}\}$ are sparse. In the problem of disease diagnosis, a patient typically has only few symptoms, and the number of possible symptoms can be several hundreds (i.e., $|x_+| \ll m$). That is, the symptom space of the disease diagnosis problem is sparse.

To formulate our problem as a decision process, we introduce the Markov decision process (MDP) terminologies used in this paper. We shall follow the standard notations in [13]. An MDP $M$ is a five tuple $M = (\mathcal{S}, \mathcal{A}, \mathcal{R}, p, \gamma)$, where: $\mathcal{S}$ is a set of states; $\mathcal{A}$ is a set of actions; $\mathcal{R} \subseteq \mathbb{R}$ is a set of possible rewards; $p : \mathcal{S} \times \mathcal{R} \times \mathcal{S} \times \mathcal{A} \to [0,1]$ is the dynamics of the MDP, with $p(s', r \mid s, a) := \Pr\{S_t = s', R_t = r \mid S_{t-1} = s, A_{t-1} = a\}$, for all $s, s' \in \mathcal{S}, r \in \mathcal{R}, a \in \mathcal{A}$; and $\gamma \in [0,1]$ is the discount-rate parameter. An MDP is said to be finite if $\mathcal{S}$, $\mathcal{A}$, and $\mathcal{R}$ are finite.

Given our dataset $\mathcal{D}$, we can construct a sample model $M = (\mathcal{S}, \mathcal{A}, \mathcal{R}, p, \gamma)$ as follows. The state space $\mathcal{S}$ is $\{-1, 0, 1\}^m$. A state can be viewed as the partial information of a patient's symptom profile $x \in \mathcal{X}$. In the state encoding, we use 1 for positive symptoms, $-1$ for negative symptoms, and 0 for unknown symptoms. The action space $\mathcal{A}$ is $\{1, \ldots, m\} \cup \{m+1, \ldots, m+n\}$. We say that an action is a *feature acquisition action* if it is less than or equal to $m$; otherwise, it is a *classification action*. The reward space $\mathcal{R}$ is $\{-1, 0, 1\}$. The dynamics $p$ is a generative model which can generate sample episodes. At the beginning, the sample model generates an initial state $S_1$ with one positive symptom by

$$(x, y) \sim \text{uniform}(\mathcal{D}), \qquad i \sim \text{uniform}(x_+), \qquad S_1 = e_i,$$

where $e_i$ is the vector in $\mathbb{R}^m$ with $i^{th}$ component 1 and all other components 0. The initial state can be viewed as the information of an initial symptom provided by a patient.

In the following time step $t$, if $M$ receives an action $A_t \leq m$ (i.e., a feature acquisition action) from the agent and the time step $t$ is less than the maximum length $T$, the sample model generates the next state $S_{t+1}$ and the reward $R_{t+1}$ by

$$S_{t+1,j} = \begin{cases} 2x_j - 1 & \text{if } j = A_t \\ S_{t,j} & \text{otherwise} \end{cases}, \qquad R_{t+1} = 0.$$

The next state $S_{t+1}$ contains the positive and negative symptoms in $S_t$, and also one additional feature $x_{A_t}$ (i.e., the $A_t^{th}$ position of $x$) acquired by the action $A_t$. If $M$ receives a feature acquisition action at the last step of an episode ($t = T$), the next state $S_{t+1}$ and the reward $R_{t+1}$ are

$$S_{t+1} = S_\perp, \qquad R_{t+1} = -1,$$

where $S_\perp$ is the terminal state. Therefore, the agent is punished if it asks too many questions. On the other hand, if $M$ receives an action $A_t > m$ (i.e., a classification action), the sample model generates $S_{t+1}$ and $R_{t+1}$ by

$$S_{t+1} = S_\perp, \qquad R_{t+1} = 2\delta_{y+m,A_t} - 1,$$

where $\delta$ is the Kronecker delta. The reward $R_{t+1}$ is 1 if the disease prediction made by the agent is correct (i.e., $y + m = A_t$). Otherwise, $R_{t+1}$ is $-1$ for a wrong disease prediction ($y + m \neq A_t$). The next state $S_{t+1}$ is the terminal state $S_\perp$, and the episode terminates.

In reinforcement learning, a policy $\pi : \mathcal{S} \to \mathcal{A}$ is a strategy function of an agent that is used to interact with an MDP. At time step $t$, the agent chooses an action based on $\pi$ and obtains an immediate reward $R_{t+1}$ generated by the MDP. The cumulative discounted reward $G_t$ obtained by the agent is $G_t = \sum_{t'=t}^{T} \gamma^{t'-t} R_{t+1}$. Given a policy $\pi$, the state value of state $s$ under policy $\pi$ is defined as $v_\pi(s) := E_\pi[G_t \mid S_t = s]$, and the action value of taking action $a$ in state $s$ under policy $\pi$ is defined as $q_\pi(s,a) := E_\pi[G_t \mid S_t = s, A_t = a]$. Therefore, the optimal action value is defined as $q_*(s,a) := \sup_\pi q_\pi(s,a)$. For a finite MDP, the optimal policy is defined as $\pi_*(s) = \arg\max_{a \in \mathcal{A}} q_*(s,a)$.

We use the policy-based method REINFORCE [17] to learn the optimal policy $\pi_*$, which goal is to maximize the cumulative discounted reward $G_t$. To achieve "fast" disease diagnosis, we have two candidate approaches. The first approach is to penalize the agent with a negative reward whenever the agent queries a symptom. Since the agent can easily learn unexpected behavior if the reward criteria is slightly changed, the severity of penalty is difficult to determine. The other approach is to set the discount-rate parameter $\gamma < 1$. The discount-rate parameter $\gamma$ can affect the cumulative discounted reward $G_t$, and simultaneously affect the strategy of the agent. If $\gamma < 1$ and the agent makes a correct prediction at the end, the agent asking fewer questions receives more cumulative discounted reward $G_t$ than that asking more questions. Therefore, the agent shall learn to shorten the diagnosis process. This approach is preferable because its reward (penalty) setting is simpler.

## 3    Our Proposed Schemes

As illustrated in Section 2, the feature vector of symptoms of each training instance is sparse in our disease diagnosis problem. If the agent relies on sampling randomly in the symptom space to query a patient, it is with low probability that a symptom probing would be responded positively by the patient. Without any symptoms being positive, the agent cannot perform disease prediction. Therefore, it is evident that in the early phrase of symptom probing, the agent must employ a strategy that can quickly find some positive symptoms during reinforcement learning. However, the naïve reinforcement learning algorithm presented in Section 2 does not explicitly guide the agent to discover positive symptoms quickly, which makes the learning progress slow under the challenging scenario of sparse features. In this section, we propose two novel techniques to guide the reinforcement learning agent to quickly discover key positive symptoms. The first technique is built on the foundation of reward shaping to encourage discovering *positive* symptoms; the second technique leverages the concept of representation learning to ensure focusing on *key* positive symptoms. As we will show in Section 4, these two techniques significantly improve the REINFORCE framework in both speed and accuracy.

### 3.1    Exploring Positive Symptoms with Reward Shaping

In order to encourage the reinforcement learning agent to discover positive symptoms more quickly, a simple heuristic is to provide the agent with auxiliary piece of reward when a positive symptom is queried, and a relatively smaller (or even negative) reward when a negative symptom is queried. However, this heuristic suffers from the risk of changing the optimal policy of the original MDP. We first introduce a principled framework that allows changing the reward function while keeping the optimal policy untouched. We then design a novel reward function that satisfies both the intuition behind the heuristic and the requirement of the framework.

The principled framework is called reward shaping, which changes the original reward function of an MDP into a new one in order to make the reinforcement learning problem easier to solve [11]. We consider the special case of potential-based reward shaping, which comes with theoretical guarantees on the invariance of the optimal policy [9]. Given an MDP $M = (\mathcal{S}, \mathcal{A}, \mathcal{R}, p, \gamma)$, potential-based reward shaping assigns a potential on each state by a bounded potential function $\varphi \colon \mathcal{S} \to \mathbb{R}$. Physically, the potential function represents the prior "goodness" of each state. We define the auxiliary reward function from state $s$ to state $s'$ as $f(s, s') := \gamma\varphi(s') - \varphi(s)$, which carries the discounted potential difference between two states. Potential-based reward shaping considers a new MDP $M_\varphi = (\mathcal{S}, \mathcal{A}, \mathcal{R}_\varphi, p_\varphi, \gamma)$ with the changed reward distribution $p_\varphi$ to be

$$p_\varphi(s', r \mid s, a) := p(s', r - f(s, s') \mid s, a), \forall s, s' \in \mathcal{S}, r \in \mathcal{R}_\varphi, a \in \mathcal{A},$$
$$\text{where } \mathcal{R}_\varphi := \{r + f(s, s') : r \in \mathcal{R}, s, s' \in \mathcal{S}\}$$

That is, the new MDP provides an additional reward $f(s, s')$ when the agent moves from state $s$ to state $s'$. The following theorem establishes the guarantee that the optimal policy of $M$ and $M_\varphi$ are equivalent.

**Theorem 1** (Policy Invariance Theorem [9]). *For any given MDP $M = (\mathcal{S}, \mathcal{A}, \mathcal{R}, p, \gamma)$ and any potential function $\varphi$, define the changed MDP $M_\varphi = (\mathcal{S}, \mathcal{A}, \mathcal{R}_\varphi, p_\varphi, \gamma)$ above. Then, every optimal policy in $M_\varphi$ is also optimal in $M$.*

With Theorem 1 guaranteeing the invariance of the optimal policies, the only issue left is to design a proper potential function $\varphi(s)$ for each state $s$. Recall that the potential function needs to match the heuristic of providing a more positive reward after identifying a positive symptom. Next, we propose a simple potential function, and then prove a necessary condition for the potential function to match the heuristic.

The simple potential function is defined as

$$\varphi(s) := \begin{cases} \lambda \times |\{j \colon s_j = 1\}| & \text{if } s \in \mathcal{S} \setminus \{S_\perp\} \\ 0 & \text{otherwise} \end{cases}, \tag{1}$$

where $\lambda > 0$ is a hyperparameter that controls the magnitude of reward shaping. That is, when the state $s$ is not the terminal state $S_\perp$, $\varphi(s)$ carries a scaled version of the number of positive symptoms that have been identified before entering state $s$. By the definition $f(s, s') := \gamma\varphi(s') - \varphi(s)$, scaling up $\lambda$ would be equivalent to scaling up $f$ to make the auxiliary rewards larger in magnitude. Once the agent reaches the terminal state $S_\perp$, the potential value of it is changed to $0$ to ensure the policy invariance [9].

After defining the potential function, we would like to ensure that the auxiliary reward $f(s, s')$ satisfies our heuristic. The heuristic is to provide the agent with a positive auxiliary reward when it discovers a positive symptom, and a negative auxiliary reward when it queries a negative symptom. Recall that we set the discount-rate parameter $\gamma < 1$ to shorten the diagnosis process. Since the discount-rate parameter $\gamma$ can affect the auxiliary reward $f(s, s')$, we need to investigate the range of $\gamma$ to make the auxiliary reward satisfy our heuristic. A similar analysis is performed by Grzes and Kudenko [1]. Here, we provide a more rigorous analysis. For any state $s$, consider a state $s_+$ where $s_+$ and $s$ differ by one positive symptom. Similarly, consider a state $s_-$ where $s_-$ and $s$ differ by one negative symptom. The following theorem establishes the condition on $\gamma$ that makes $f(s, s_+)$ positive and $f(s, s_-)$ non-positive.

**Theorem 2.** *Let $\varphi \colon \mathcal{S} \to \mathbb{R}_{\geq 0}$ be a bounded function over $\mathcal{S}$, $u = \sup\{\varphi(s) : s \in \mathcal{S}\}$. If $s, s_+, s_- \in \mathcal{S}$ satisfies $\varphi(s_+) = \varphi(s) + c$ and $\varphi(s_-) = \varphi(s)$ for some $c > 0$, then, for any $\frac{u}{u+c} < \gamma < 1$,*

$$f(s, s_-) \leq 0 \text{ and } f(s, s_+) > 0.$$

*Proof.* $f(s, s_-) = (\gamma - 1)\varphi(s)$ which is trivially non-positive by the conditions $\gamma < 1$ and $\varphi(s) \geq 0$; $f(s, s_+) = \gamma\varphi(s_+) - \varphi(s) > \left(\frac{u}{u+c}\right)(\varphi(s) + c) - \varphi(s) \geq \left(\frac{\varphi(s)}{\varphi(s)+c}\right)(\varphi(s) + c) - \varphi(s) \geq 0.$ $\square$

The proposed potential function in (1) easily satisfies the condition of Theorem 2 by setting $c = \lambda$ and $u = \lambda d$, where $d = \max\{|x_+| : x \in \mathcal{D}_\mathcal{X}\}$ is the sparsity level of dataset $\mathcal{D}$. Then, by choosing $\gamma$ to be within the range of $(\frac{\lambda d}{\lambda d + \lambda}, 1) = (\frac{d}{d+1}, 1)$, we can ensure that the auxiliary reward of discovering a positive symptom to be positive, and the auxiliary reward of discovering a negative symptom to

be non-positive, and therefore, satisfying the simple heuristic under the principled framework of potential-based reward shaping.

Given that certain important negative symptoms are also helpful to distinguish diseases, one may think it is counterintuitive to punish an agent with non-positive auxiliary rewards when it queries negative symptoms. However, Theorem 1 guarantees the invariance of the optimal policy with auxiliary rewards. Therefore, even though the agent may receive non-positive auxiliary rewards, it can still learn to query those critical negative symptoms.

## 3.2 Feature Rebuilding of Sparse Features

One specialty of the disease diagnosis problem is that many symptoms can be highly correlated. Symptom correlation allows doctors to use some positive symptoms to infer the existence of some other symptoms. For example, a patient suffers from runny nose may also suffer from sore throat or headache. Our agent can take advantage of such correlation without "wasting" queries on probing those highly correlated symptoms.

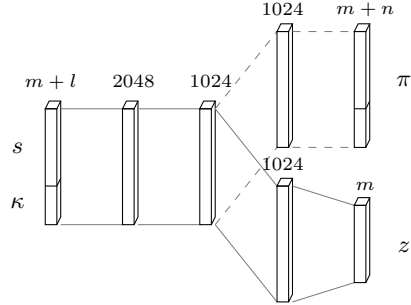

Figure 1: Dual neural network architecture. The upper branch is the policy $\pi$ of an agent. The lower branch is the feature rebuilding part of sparse features.

The MDP designed in Section 2 allows the agent to learn to not waste queries by encouraging the agent to obtain the correct classification reward as quickly as possible. Nevertheless, the reward signal, which comes at the moment of the final action of classification, is rather weak to the agent. Thus, the concept of not wasting queries can be slow to learn. Furthermore, the reward signal is weakened by the technique of reward shaping introduced previously. In particular, because any positive symptom would receive the same amount of auxiliary reward from the changed MDP, the agent is generally more eager to identify *any* positive symptom (to receive some auxiliary reward immediately) rather than a *key* positive symptom that can help infer other positive symptoms towards a quicker classification reward.

To identify key positive symptoms, we combine a symptom rebuilding task with policy learning. Figure 1 illustrates our proposed architecture: the first three layers are *shared layers*, the upper branch is for *policy learning*, and the lower branch is for *feature rebuilding*. In the training stage, the model should generate a policy $\pi$, as well as rebuild full symptoms from partial ones. If the agent queries more key positive symptoms, the model can make a better feature rebuilding at the next time step. A better feature rebuilding provides the agent a preferable representation in the shared layers. That is, the agent can possess a global view of the symptom information. Referring this global information, the agent inclines to make a correct diagnosis and to receive a positive reward. Afterwards, the desired behavior of querying key positive symptoms will be reinforced by the positive reward.

As shown in Figure 1, the input to the neural network is the concatenation of the state $s$ and contextual information $\kappa \in \{0,1\}^l$. We use the same contextual information $\kappa$ as in [3]. Let $x \in \mathcal{X}$ be a vector with full symptom information and $z \in \mathbb{R}^m$ the rebuilt one calculated by the lower branch (with a sigmoid output function). We propose to consider the binary cross-entropy function [16] as the feature rebuilding loss:

$$J_{reb}(x, z; \theta) = -\sum_j \sum_i [x_i^{(j)} \log z_i^{(j)} + (1 - x_i^{(j)}) \log(1 - z_i^{(j)})]. \tag{2}$$

We propose to minimize the feature rebuilding loss together with maximizing the cumulative discounted reward in REINFORCE. In particular, consider an objective function

$$J = J_{pg}(\theta) - \beta J_{reb}(x, z; \theta).$$

The first term, $J_{pg}(\theta) = v_{\pi_\theta}(s_0)$, is the cumulative discounted reward from the initial state to the terminal state. The second term is feature rebuilding loss (i.e., Equation 2) and $\beta$ is a hyperparameter, which controls the importance of the feature rebuilding task.

Optimizing the new objective function requires some simple changes of the original stochastic gradient ascent algorithm within REINFORCE. In particular, the policy gradient theorem already

---

**Algorithm 1:** REFUEL: Exploring sparse features using reward shaping and feature rebuilding

**Input** : A dataset $\mathcal{D} = \left\{ \left( x^{(j)}, y^{(j)} \right) : x^{(j)} \in \mathcal{X}, y^{(j)} \in \mathcal{Y} \right\}_{j=1}^{k}$, where $\mathcal{X} = \{0,1\}^m$ and $\mathcal{Y} = \{1, \dots, n\}$. An action set
$\mathcal{A} = \{1, \dots, m\} \cup \{m+1, \dots, m+n\}$.
$N$ is the number of episodes. $T$ is the maximum number of steps of an episode.

**Output** : Agent's parameters $\theta$

```
1  begin
2      Initialize parameters θ
3      for l ← 0 to N do
4          t ← 1
5          (x, y) ~ uniform(D), x₊ := {j : xⱼ = 1}, i ~ uniform(x₊)
```

6 $\quad\quad S_{t,j} \longleftarrow \begin{cases} 1 & \text{if } j = i \\ 0 & \text{otherwise} \end{cases}$

```
7          // start one sample episode
8          repeat
9              Aₜ ~ π(Aₜ | Sₜ);
10             if Aₜ ≤ m and t < T then
```

11 $\quad\quad\quad\quad S_{t+1,j} \longleftarrow \begin{cases} 2x_j - 1 & \text{if } j = A_t \\ S_{t,j} & \text{otherwise} \end{cases} \; ; \; R_{t+1} \longleftarrow 0$

```
12             else if Aₜ ≤ m and t ≥ T then
13                 S_{t+1} ← S⊥ ; R_{t+1} ← −1
14             else
15                 S_{t+1} ← S⊥ ; R_{t+1} ← 2δ_{y+m,Aₜ} − 1
16             end
17             t ← t + 1
18         until Sₜ = S⊥
19         // REFUEL: reward shaping and feature rebuilding
20         φ(s) := λ × |{j : sⱼ = 1}|
21         Gₜ ← 0; Δθ ← 0; t ← t − 1
22         repeat
23             Gₜ ← R_{t+1} + (γφ(S_{t+1}) − φ(Sₜ)) + γG_{t+1}
24             Δθ ← Δθ + Gₜ (∇_θ π(Aₜ|Sₜ,θ))/(π(Aₜ|Sₜ,θ)) + β∇J_{reb}(x, zₜ; θ) + η∇H(π(Aₜ|Sₜ,θ))
25             t ← t − 1
26         until t = 0
27         θ ← θ + αΔθ
28     end
29     return θ
30 end
```

---

establishes that $\nabla J_{pg}(\theta) \propto \mathbb{E}_\pi \left[ G_t \frac{\nabla_\theta \pi(A_t|S_t,\theta)}{\pi(A_t|S_t,\theta)} \right]$. Thus, the update rule for the new objective function is simply

$$\theta' = \theta + \alpha \left[ G_t \frac{\nabla_\theta \pi(A_t|S_t,\theta)}{\pi(A_t|S_t,\theta)} + \beta \nabla J_{reb}(x, z_t; \theta) + \eta \nabla H(\pi(A_t|S_t,\theta)) \right].$$

Note that we follow a popular practice of adding an additional entropy regularization term $H$, which was first proposed by [18] to help the agent escape from local optimal at the beginning of the training process.

By additionally minimizing the feature rebuilding loss, we direct the agent to internally carry the knowledge towards good feature rebuilding (low rebuilding loss), and the knowledge can naturally be used to guide a confident classification action more quickly.

### 3.3 The REFUEL Algorithm

Combining the techniques of reward shaping and enforcing feature rebuilding, we propose the REFUEL algorithm to solve the disease diagnosis problem, as shown in Algorithm 1. The inputs to the algorithm include a training dataset $\mathcal{D}$, an action set $\mathcal{A}$, the number of episodes $N$, and the maximum number of steps of an episode $T$. The output of the algorithm is the trained model.

At the beginning of the training process, the algorithm initializes the model's parameters and enters a training loop. The sample model generates an initial state with a positive symptom to simulate a patient with an initial symptom (lines 5 - 6). Afterwards, the algorithm starts an episode. Within the episode, the agent selects an action based on the current policy $\pi$. Once the agent completes an action, the immediate reward $R_t$ and the state $S_t$ are stored in the buffer for further use by the reward shaping technique.

When the agent makes a decision, two types of actions can be performed. If the agent selects an acquisition action ($A_t \leq m$), the sample model generates the next state, which consists of the symptoms from the previous state and one additional symptom acquired by the action, and a reward 0 for $t < T$ (line 11), or the terminal state $S_\perp$ and a reward $-1$ for $t \geq T$ (line 13). If the agent selects a classification action ($A_t > m$), it receives a reward 1 for a correct prediction; otherwise, the agent receives a reward $-1$. Subsequently, the agent reaches the terminal state $S_\perp$ and the episode terminates (line 15).

Once the episode terminates, the algorithm takes the immediate rewards $R_t$ and the states $S_t$ stored in the buffer to calculate shaped rewards that encourage the identification of positive symptoms (line 23). Then, the algorithm takes the rebuilt symptoms $z_t$ to compute the feature rebuilding loss, and uses the update rule of the new objective function to update model parameters (line 24).

## 4  Experiments

Owing to the privacy laws (e.g., the Health Insurance Portability and Accountability Act; HIPAA), it is difficult to obtain real-world medical data at the current points. We follow our earlier work on the simulation procedure to generate artificial medical data instead [3]. The procedure maintains a probability distribution of a patient's symptoms given the patient has a certain disease. The probability distribution is over 650 diseases and 376 symptoms. The simulation procedure first uniformly samples a disease from a disease set. Under the sampled disease, the procedure extracts the probabilities of symptoms given the sampled disease. Then, the procedure generates the symptoms of the sampled disease by performing a Bernoulli trial on each symptom. For example, if a disease called common cold is sampled by the procedure, the probabilities of cough and sore throat under common cold (which are 85% and 82%) can be extracted. Then doing Bernoulli trials according to these probabilities can generate one instance.

In all of our experiments, we used the simulation procedure [3] to sample $10^6$, $10^4$, $10^5$ data for training, validation, and testing. We used the same neural network architecture (Figure 1) and the same hyperparameters in all experiments. We used Adam [4] as our optimizer. The initial learning rate is $0.0001$, and the batch size is $512$. The coefficient of the entropy regularizer $\eta$ is initially set to $0.01$ and linearly decayed to 0 over time. The coefficient of the feature rebuilding loss $\beta$ is 10. The discount-rate parameter $\gamma$ is 0.99, and the hyperparameter $\lambda$ in our potential function $\varphi$ is 0.25. We report the sensitivity analyses of hyperparameters $\beta$ and $\lambda$ in the supplementary material.

In our implementation, we changed the auxiliary reward $f$ in the terminal state. Specifically, when the agent encounters a terminal state $S_\perp$, it receives an auxiliary reward $f(S, S_\perp) = \gamma \varphi(S_\perp)$ instead of $f(S, S_\perp) = \gamma \varphi(S_\perp) - \varphi(S)$. We found that this deviation from the standard reward shaping technique resulted in better performance. Our hypothesis is that the modified auxiliary reward $f$ can encourage the agent to query more positive symptoms because our definition of $\varphi$ actually counts the number of positive symptoms.

To evaluate the learning efficiency and accuracy of our REFUEL algorithm, we formed three different disease diagnosis tasks, which consist of top-200, top-300, and top-400 diseases in terms of the number of occurrences in the Centers for Disease Control and Prevention (CDC) records. Figure 2 shows the experimental results. The baseline (blue line) is REINFORCE [17] without any enhancement; RESHAPE (yellow line) is the baseline plus the technique of reward shaping; REFUEL (red line) is the baseline plus both reward shaping and feature rebuilding. Each line in Figure 2 is the average training accuracy over 5 different random seeds. The shaded area represents the region of two standard deviations. Note that as shown in Figure 2, RESHAPE and REFUEL outperform the baseline, especially in the tasks of 300 and 400 diseases. We can also see that REFUEL can explore faster and predict better than RESHAPE.

Figure 3 shows the comparison of cumulative discounted rewards received in the training process. Since the reward and accuracy are highly correlated, the trends of these two figures are similar. Also, we can see that in the case of 400 diseases, the baseline struggles for obtaining positive rewards. Both RESHAPE and REFUEL can obtain rewards far more than the baseline. This suggests that our algorithm is effective in discovering key positive symptoms.

Since in our reward setting, there is no penalty for the agent if it selects a repeated action. We are interested in seeing whether our algorithm can more robustly avoid repeated actions. Figure 4 shows the probability of choosing an repeated action in each step. The curves in Figure 4 are much more different from the curves in Figure 2 and 3. While the repeat rate of the baseline sharply increases at

the end of the training process in the cases of 300 and 400 diseases, RESHAPE and REFUEL can quickly converge towards 0.

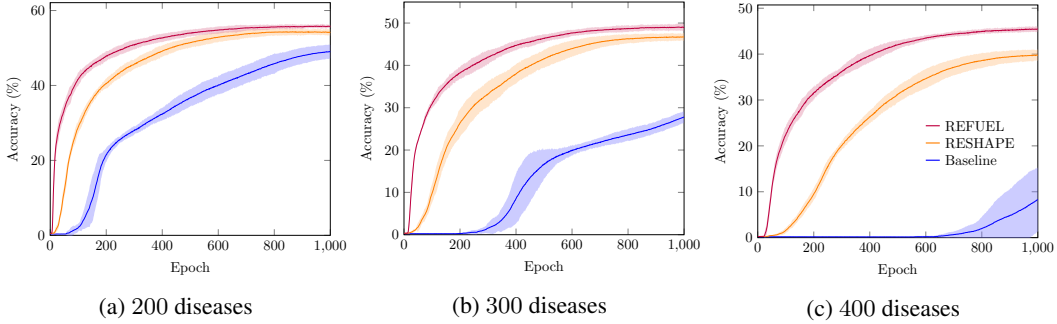

(a) 200 diseases        (b) 300 diseases        (c) 400 diseases

Figure 2: Experiments on 3 datasets of different disease numbers. The curves show the training accuracy of three methods. REFUEL (red line) uses reward shaping and feature rebuilding; RESHAPE (yellow line) only uses reward shaping; Baseline (blue line) adopts none of them. The solid line is the averaged result of 5 different random seeds. The shaded area represents two standard deviations.

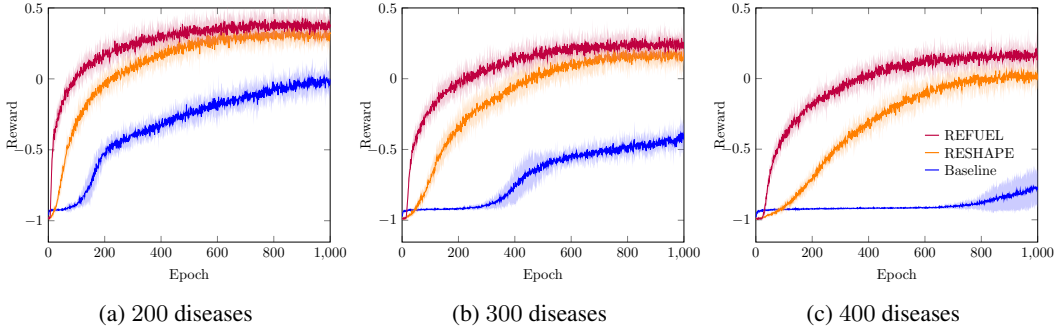

(a) 200 diseases        (b) 300 diseases        (c) 400 diseases

Figure 3: Experiments on 3 datasets of different disease numbers. Each plot shows the average of cumulative discounted rewards during training.

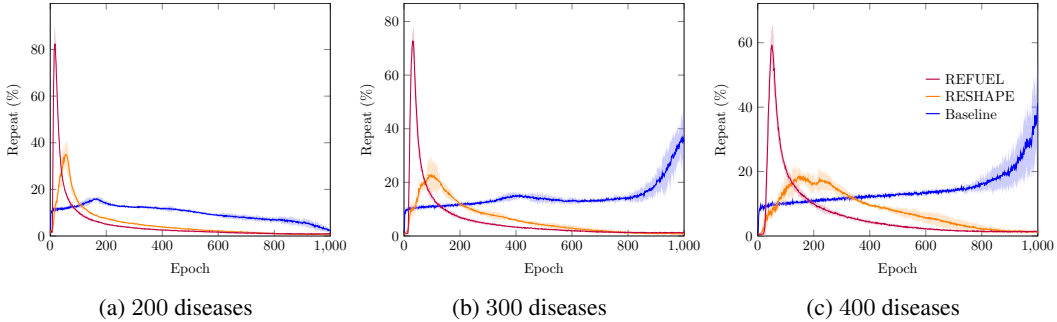

(a) 200 diseases        (b) 300 diseases        (c) 400 diseases

Figure 4: The probability of choosing a repeated action during training for each dataset.

After training, the models were selected according to the performance on validation set. The testing results are reported in Table 1. The table clearly shows that all the results from REFUEL outperform the baseline. Note that the performance of the baseline in the case of 400 diseases is only slightly better than a random guess, whereas REFUEL can obtain a top-5 accuracy of 67.09% in 400 diseases.

Table 2 shows the statistics of symptom acquisition. In each case, the number of possible symptoms is more than 300, and the average number of positive symptoms per patient is only about 3. Furthermore, there is 1 positive symptom in the initial state. That is, the average maximum number of queriable positive symptoms is only about 2. Therefore, it is extremely hard to find the positive symptoms. As showed in Table 2, the baseline can only acquire 0.01 positive symptoms on average in the case of 400 diseases. On the contrary, our proposed REFUEL algorithm can acquire 1.31 positive symptoms. Thus, this experiment indicates that our algorithm is beneficial for sparse feature exploration.

Table 1: The test accuracy of baseline and REFUEL.

| #Diseases | Baseline | | | | REFUEL | | | |
|---|---|---|---|---|---|---|---|---|
| | Top 1 | Top 3 | Top 5 | #Steps | Top 1 | Top 3 | Top 5 | #Steps |
| 200 | 48.75 | 61.70 | 66.55 | 6.66 | **54.84** | **73.66** | **79.68** | 8.01 |
| 300 | 21.78 | 28.42 | 31.18 | 6.78 | **47.49** | **65.08** | **71.17** | 8.09 |
| 400 | 0.74 | 1.46 | 2.09 | 8.98 | **43.83** | **60.76** | **67.09** | 8.35 |

Table 2: The average number of positive symptoms that the agent successfully inquires.

| #Diseases | #Possible Symptoms | #Symptoms / Patient | Baseline | REFUEL |
|---|---|---|---|---|
| 200 | 300 | 3.07 | 0.56 | **1.40** |
| 300 | 344 | 3.12 | 0.33 | **1.37** |
| 400 | 354 | 3.19 | 0.01 | **1.31** |

Table 3: Comparison of REFUEL with the best prior work published in AAAI 2018 [3].

| #Diseases | Hierarchical Model [3] | | | | REFUEL | | | |
|---|---|---|---|---|---|---|---|---|
| | Top 1 | Top 3 | Top 5 | #Steps | Top 1 | Top 3 | Top 5 | #Steps |
| 73 | 63.55 | 73.35 | 77.94 | 7.15 | **69.15** | **86.85** | **91.71** | 7.52 |
| 136 | 44.50 | 51.90 | 55.03 | 5.73 | **60.60** | **79.57** | **84.98** | 7.70 |
| 196 | 32.87 | 38.02 | 40.20 | 5.14 | **55.01** | **74.20** | **80.22** | 7.96 |
| 255 | 26.26 | 29.81 | 31.42 | 5.01 | **50.66** | **68.96** | **75.04** | 8.14 |

Next, we compare with prior works. Since the previous study [2] reported that the RL method readily outperformed tree-based methods, we put the experimental results of a tree-based method in the supplementary material. Here, we focus on the hierarchical model proposed in [3], which is the best prior work. This prior work creates four different tasks of disease sizes 73, 136, 196, 255. We train 5 models with 5 different random seeds for each task. The average test accuracy are reported in Table 3, which reveals that REFUEL significantly outperforms the hierarchical model. The top-1 accuracy of REFUEL is nearly as twice as the hierarchical model in 255 diseases. Besides, the top-3 and top-5 accuracies are significantly boosted from top-1 accuracy in REFUEL. This is particularly helpful in the disease diagnosis problem if some diseases are very difficult to detect solely based on symptoms.

## 5 Concluding Remarks

In this paper, we presented REFUEL for disease diagnosis. The contributions of REFUEL are two folds. First, we design an informative potential function to guide an agent to productively search in the sparse feature space and adopt a reward shaping technique to ensure the policy invariance. Second, we design a component that rebuilds the feature space to provide the agent with a better representation. The two techniques help identify symptom queries that yield positive patient responses at a much higher probability, which in term leads to faster exploration speed and higher diagnosis accuracy. The experimental results have confirmed the validity of our proposed method and shown that it is a promising approach to fast and accurate disease diagnosis.

Also, we discovered that slightly modifying the auxiliary reward in the terminal state resulted in better performance. To the best of our knowledge, its theoretical justification is still open. We will further investigate this interesting finding in our future work.

We have deployed REFUEL with our DeepQuest service in more than ten hospitals. Our future work plans to enhance REFUEL to deal with continuous features so that the algorithm can be applied to some other domains with features of continuous values.

## Acknowledgments

The work was carried out during the first author's internship at HTC Research and became part of the Master's thesis of the first author [10]. We thank Profs. Shou-De Lin, Yun-Nung Chen, Min Sun and the anonymous reviewers for valuable suggestions. We also thank the members at HTC Research: Jin-Fu Lin for the support of operating GPU instances and Yang-En Chen for his efforts in organizing the experiment results. This work was partially supported by the Ministry of Science and Technology of Taiwan under MOST 107-2628-E-002-008-MY3.

## Footnotes

[1]REFUEL stands for REward shaping and FeatUrE rebuiLding.

[2]We use positive and negative symptoms to denote present and absent symptoms.

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
