[Supplementary Material]

# Supplementary Material

## 1 Sensitivity Analyses of Hyperparameters

(a) Hyperparameter $\lambda$

(b) Hyperparameter $\beta$

Figure 1: The sensitivity analyses of hyperparameters $\lambda$ and $\beta$ in the case of 400 diseases.

We perform the sensitivity analyses of hyperparameters $\lambda$ and $\beta$ in the case of 400 diseases. First, we analyze the behavior of REFUEL with varying $\lambda \in \{0, 0.125, 0.25, 0.375, 0.5\}$. Recall that we define our potential function as

$$\varphi(s) := \begin{cases} \lambda \times |\{j \colon s_j = 1\}| & \text{if } s \in \mathcal{S} \setminus \{S_\perp\} \\ 0 & \text{otherwise} \end{cases},$$

where $\lambda$ controls the magnitude of reward shaping. As shown in Figure 1a, the agent with higher $\lambda$ learns faster at the beginning, but the agent with $\lambda = 0.25$ (red line) reaches the highest accuracy at the end. Therefore, we choose $\lambda = 0.25$ as our hyperparameter.

Next, we conduct the experiments with different $\beta \in \{0, 2.5, 5, 7.5, 10\}$. In our objective function $J = J_{pg}(\theta) - \beta J_{reb}(x, z; \theta)$, the hyperparameter $\beta$ controls the importance of the feature rebuilding task $J_{reb}$. Figure 1b shows that higher $\beta$ yields better performance. Therefore, we select $\beta = 10$ as our hyperparameter.

## 2   Comparison with Decision Tree

(a) 200 diseases    (b) 300 diseases    (c) 400 diseases

Figure 2: Experiments on 3 datasets of different disease numbers.

We apply the CART decision tree algorithm to the disease diagnosis problem. In Figure 2, the red line represents the performance of REFUEL; the blue line is the result of the CART decision tree algorithm. Whereas REFUEL requires about $8$ symptoms from a patient, the CART decision tree algorithm requires about $50$ symptoms to reach the same performance as REFUEL, which is impractical to the disease diagnosis problem.