[Reviews · NeurIPS 2018]

Reviewer 1



The authors describe an RL architecture comprised of reward shaping plus representation learning that is used to solve an active classification problem, framed as "diagnosis." In this setting, an agent can measure the value of "symptoms" at some cost, and eventually makes a prediction of what disease is present. The architecture is intended to take advantage of the property that symptoms are sparse but correlated. Reward shaping is used to help the agent learn to quickly find symptoms that are present, while the correlations are used to avoid having the agent measure symptoms that are already "known" based on already-measured ones with high certainty. Experimental results demonstrate a substantial improvement over prior work. Quality: I think the paper addresses an interesting problem and I don't see any technical flaws. I am a bit unsure of the significance of the results because I am not sure how other non-RL methods would perform in this setting. When the authors discuss related tree-based methods, they state that "Although they design sophisticated impurity function, it is hard to obtain good results with small budgets." I think the authors either make the argument in their experimental results section that the tree-based methods are not worth competitors, or provide results for comparison. A discussion of computational burden relative to previous methods would improve the paper as well. 25 with small budgets. Clarity: I found the paper clearly presented. Suggestion: In mental health especially, the terms "positive" and "negative" w.r.t. symptoms already has a technical meaning. It may be better to go with "present" and "absent" rather than "positive" and "negative." Originality: The authors clearly identify the previous work they want to build on; however as mentioned above under "quality" it's not clear whether the experimental results signify a new advance or not. Significance: I could classify this paper as a useful and thoughtful combination of existing ideas. I think the methods presented would useful for many kinds of diagnosis problems besides the one presented. As a future work possibility, the authors may want to consider the concept of "symptoms" versus "signs" in medicine; creating more nuanced models that distinguish between these could be interesting future work. https://en.wikipedia.org/wiki/Medical_sign Specific comments: 26: "can prevent making a query in local view" - unclear what this means

Reviewer 2



Dear Authors. Thank you for your response. I still have concerns about your explanation of the discount factor. I can clarify that your policies with gamma=1 shown in rebuttal in Figures 2 and 3 don't ask more questions because your final reward (+1 for correct prediction) is too small. If the final reward was larger (you would need to tune it or just set it to a "very" large value), the agent would ask more questions without a problem. This means that gamma=1 would be sufficient to solve this problem. If the neural network requires bounding rewards, all the rewards could be scaled down, and the step penalty would be proportionally smaller than -1. As I said in my original review, your paper is a nice and well-written work, but the discount factor is problematic in this paper, and I feel that I should not argue for acceptance, although I would not be upset if the paper was accepted. If your paper was a purely applied paper, I could support it, but since you are claiming technical contributions, you need to provide convincing arguments that they are necessary. This is something that is missing in your paper. ----- A medial application is addressed using reinforcement learning. The goal of this work is to compute a policy that will detect a disease, minimising the number of features tested. The authors introduce two distinct algorithmic contributions. The first one shows how reward shaping can be applied to this domain. The second contribution is an extended neural network for the policy gradient method that predicts the unobserved features in a dedicated output layer. The experimental results on simulated data indicate that the baseline method combined with reward shaping and learning with additional output layer lead to the best performance with respect to both convergence speed and the quality of the final policy. This paper presents good research, but I am not sure if it can be accepted in this form. Writing is very good in general, but the introduction fails to provide a clear problem statement. For example, the term "query budget" is not explained in the introduction, but budgeted learning in machine learning often assumes fixed budget, and it is not the case in this paper. A precise definition of budget is required because now, after reading section 1, it is not clear what the real, technical challenge is. The authors cite tree-based greedy methods. I believe that to demonstrate benefits of reinforcement learning, it would be useful if the authors could add results of such approaches. Since the most simple baseline is quite weak, I would argue that the greedy methods can beat it. Between lines 71 and 72, the uniform distribution is used twice for the initial state. What is the reason for that? Does it mean that the initial state is randomly sampled? If so, this is probably to deal with the exploration problem? Considering the existing reinforcement learning literature, the reward defined in lines 79-80 is very unusual. To punish the agent for asking too many questions, reward of -1 should be given for every query. With this, the problem becomes a standard stochastic-shortest path problem. With that, the analysis in Fig. 4 would not be required because the alg. would be penalised for repeated queries. The dual neural network shown in Fig. 1 is interesting and it is in line with the current developments in various applications of deep learning. This is probably the strongest contribution in this paper. The authors were quite clever here, and intuitively, this idea makes a lot of sense in this application. The authors could provide a better explanation of this design because I am pretty sure that people who have not seen this trick in other deep learning papers will be confused. Theorem 2 indicates that the authors were clever observing the relationship between the discount factor and the absolute values of the potential function. However, this problem was previously discussed in "Marek Grzes, Daniel Kudenko: Theoretical and Empirical Analysis of Reward Shaping in Reinforcement Learning. ICMLA 2009: 337-344". It seems to me that the authors have a more general approach here than the past reference. Even though Theorem 2 and the past work on the same problem show an important problem related to the discount factor, I don't think that the authors of this paper have to worry about this problem. The authors could set the discount factor gamma to 1, and then the entire investigation in Theorem 2 would not be necessary. Right now, Theorem 2 can be used to determine the discount factor (which is a bit weird because Rich Sutton would surely argue that the discount factor is part of the domain), so why not to set it to the value of 1. The problem is episodic so the value of 1 would be justified or even preferred. This particular observation makes me believe that Theorem 2 represents a contribution that is not fully justified in this particular paper. The potential function used in this work is informative, and I am not surprised that reward shaping leads to better performance. This model assumes that every patient suffers from at most one disease. If two diseases were present, the algorithm would identify incorrect correlations. The authors should explain that in the paper. In Tab. 1 and 3, the baseline has fewer steps than the RE^2. Why is that? This is probably related to my comment about stochastic-shortest paths that is above. The paper is interesting, but I am not sure if the presentation and contributions are good enough to guarantee acceptance.

Reviewer 3



This is a clear well-written paper on improving interactive diagnosis systems using reinforcement learning. The two main contributions are proposing a specific form of reward shaping for interactive self-diagnosis systems, and adding a reconstruction loss in order to better deal with sparse feature spaces. Their method is evaluated on a synthetic benchmark problem used in [2] (AAAI 2018 paper) and shows significantly improved diagnosis accuracy compared to [2], as well as compared to vanilla REINFORCE. From what I can tell the paper is novel but I am not very familiar with work on diagnosis systems. It is not necessarily groundbreaking but a fine contribution that is sufficient for NIPS presentation. Some notes / suggestions for improvement: Discuss how to deal with the situation when a patient is uncertain whether he / she has a symptom or not, or where the symptoms are categorical or real-valued. Discuss sensitivity to the beta parameter, and how it could be determined given limited interaction time. Make the choice of encoding consistent - for x_j a value of zero means no symptom, whereas in the state space it means that the symptom is unknown.